# Prevalence of Chronic Kidney Disease and Variation of Its Risk Factors by the Regions in Okayama Prefecture

**DOI:** 10.3390/jpm12010097

**Published:** 2022-01-12

**Authors:** Ryoko Umebayashi, Haruhito Adam Uchida, Natsumi Matsuoka-Uchiyama, Hitoshi Sugiyama, Jun Wada

**Affiliations:** 1Department of Nephrology, Rheumatology, Endocrinology and Metabolism, Graduate School of Medicine, Dentistry and Pharmaceutical Sciences, Okayama University, Okayama 700-8530, Japan; ryoko_ume@hotmail.co.jp (R.U.); natsumi-matsuoka-uchiyama@s.okayama-u.ac.jp (N.M.-U.); hitoshis@okayama-u.ac.jp (H.S.); junwada@okayama-u.ac.jp (J.W.); 2Department of Chronic Kidney Disease and Cardiovascular Disease, Graduate School of Medicine, Dentistry and Pharmaceutical Sciences, Okayama University, Okayama 700-8530, Japan; 3Department of Human Resource Development of Dialysis Therapy for Kidney Disease, Okayama University Graduate School of Medicine, Dentistry, and Pharmaceutical Science, Okayama 700-8530, Japan

**Keywords:** chronic kidney disease, medical checkup, risk factor

## Abstract

Objective: The prevention of chronic kidney disease (CKD) progression is an important issue from health and financial perspectives. We conducted a single-year cross-sectional study to clarify the prevalence of CKD and its risk factors along with variations in these factors among five medical regions in Okayama Prefecture, Japan. Methods and Results: Data concerning the renal function and proteinuria as well as other CKD risk factors were obtained from the database of the Japanese National Health Insurance. The proportion of CKD patients at an increased risk of progression to end-stage renal disease (ESRD), classified as orange and red on the CKD heatmap, ranged from 6–9% and did not vary significantly by the regions. However, the causes of the increased severity differed between regions where renal dysfunction was predominant and regions where there were many patients with proteinuria. CKD risk factors, such as diabetes mellitus, hypertension, hyper low-density lipoprotein-cholesterolemia, obesity, smoking and lack of exercise, also differed among these regions, suggesting that different regions need tailored interventions that suit the characteristics of the region, such as an increased health checkup ratio, dietary guidance and promotion of exercise opportunities. Conclusions: Approximately 6–9% of people are at an increased risk of developing ESRD (orange or red on a CKD heatmap) among the population with National Health Insurance in Okayama Prefecture. The underlying health problems that cause CKD may differ among the regions. Thus, it is necessary to consider intervention methods for preventing CKD progression that are tailored to each region’s health problems.

## 1. Introduction

The increase in patients with chronic kidney disease (CKD) is an emerging health problem of global interest [1]. The main risk factors for developing CKD are diabetes mellitus, high blood pressure, cardiovascular disease, smoking, obesity, older age and a family history of kidney disease. In Japan, the estimated prevalence of CKD is 13% [2], and the number of chronic dialysis patients reached 2640 per 1 million population in 2017 and continues to increase annually [3]. Diabetic nephropathy is the most common cause of ESRD, thus various preventive measures against diabetic nephropathy have been taken and the number of dialysis introduction due to diabetic nephropathy has begun to decrease in Japan [3]. On the other hand, the number of patients who started renal replacement therapy is increasing year by year, therefore, the comprehensive strategies for preventing CKD development and progression are required. Furthermore, the medical cost associated with managing dialysis patients is about 4 million yen per year, and the increase in patients with end-stage renal disease (ESRD) is also an important issue in Japan. In 2019, the Japanese government requested each prefecture implement strategies to prevent the development and progression of CKD.

Okayama Prefecture is located at the western part of Japan; its southern part faces the sea with a warm climate, while the northern part has highlands and snow in the winter and a relatively cold climate. Okayama Prefecture has a population of 1.9 million and is divided into five medical regions. The southern part is divided into eastern and western parts (South-East, South-West), and the northern part is divided into an eastern part (Tsuyama-Aida), a western part (Niimi-Takahashi) and a central part (Maniwa). Okayama Prefecture has already advanced many projects, like investigating the number of CKD and chronic hemodialysis patients and their associated medical costs, and established a medical cooperation system. 

We herein report the prevalence of CKD and the associated risk factors as well as the differences in these parameters among the medical regions in Okayama Prefecture, Japan, in order to establish a strategic basis to prevent CKD progression.

## 2. Materials and Methods

Study population: National Health Insurance (NHI) in Japan mainly covers self-employed individuals, retirees and their non-working dependents under 75 years old. In Okayama Prefecture, 20.1% of the population use NHI. The NHI of each municipality performs health checkups annually for enrollees 40–74 years old, and the results are aggregated in the NIH database (Kokuho Data Base, KDB) system developed by the All-Japan Federation of NHI Organizations. 

Study design: We conducted a single-year cross-sectional study in Okayama Prefecture in 2019, using the data extracted from the KDB system. Okayama Prefecture is divided into five medical regions: South-East, South-West, Takahashi-Niimi, Maniwa, Tsuyama-Aida. We compared the incidence of a high-risk population of ESRD and the associated risk factors. The severity of CKD was assessed by CKD heatmap staging based on the renal function and degree of proteinuria, created for Japanese people from the KDOQI ESRD risk classification [2,4]. As a brief overview, the category with odds ratios of developing ESRD ranging from 1–5 was colored green, 6–20 yellow, 21–100 orange and over 100 red. We defined the risk factors for ESRD based on the HbA1c value associated with diabetes mellitus (≥6%), the systolic blood pressure value associated with hypertension (≥140 mmHg), the low-density lipoprotein (LDL)-cholesterol value associated with hypercholesteremia (≥140 mg/dL), the body mass index (BMI) associated with obesity (≥25 kg/m^2^), a smoking habit and a lack of daily exercise (exercises < 30 min/day).

Ethical statement: this study was approved by the Okayama University Institutional Review Board (No 2104-006) and was conducted in accordance with the Declaration of Helsinki principles. 

Statistical analyses: All data are presented as the mean ± standard deviation or number (%). Statistical significance among groups was assessed by a one-way analysis of variance followed by the Student–Newmann method and chi-square test using the SigmaPlot 12.5 software program (Systat Software Inc., San Jose, CA, USA). A *p*-value < 0.05 was considered to be statistically significant.

## 3. Results

### 3.1. Participants and Their Characteristics

A total of 88,420 individuals underwent NHI’s health examinations in Okayama Prefecture in 2019. The number of individuals who underwent health checkups is listed in Table 1, and the rate in each medical region ranged from 20% to 30%. There were more females than males in all groups. Individuals from the South-East region were significantly younger compared to other regions, but the mean difference was only one year (Table 1 and Appendix A).

### 3.2. Population Distribution on a CKD Heatmap

The CKD heatmaps of each medical region are shown in Figure 1. The South-East area had the highest proportion of orange and red on the CKD heat map (8.2% in Table 1), while values were lower in the Takahashi-Niimi region and Tsuyama-Aida region (6.6% and 6.7%, respectively). In the South-East region, the mean estimated glomerular filtration rate (eGFR) was low, and the ratio of people with proteinuria was higher than in other regions, indicating that this area included many people at risk of ESRD on the CKD heat map. In the Maniwa region, which had the next highest proportion of severe CKD on the heatmap (7.5%), the mean eGFR was high, as was the ratio of people with urinary proteinuria. Both the Takahashi-Niimi and Tsuyama-Aida regions had a low severity on the CKD heat map (red and orange areas were 6.6% and 6.7%, respectively). The Takahashi-Niimi region had a lower mean eGFR than other regions but a low ratio of people with proteinuria, while the Tsuyama-Aida region had a higher eGFR than other regions, but the ratio of people with proteinuria was intermediate. 

### 3.3. Differences in Risk Factors for CKD among Five Regions and Their CKD Prevention Strategies

Table 2 shows the number and ratios of risk factors for CKD, such as diabetes mellitus (HbA1c ≥ 6.0%) and diabetes mellitus at high risk of developing CKD (HbA1c ≥ 7.0%), hypertension (systolic blood pressure ≥ 140 mmHg), obesity (BMI ≥ 25 kg/m^2^), hyper LDL cholesterolemia (LDL-C ≥ 140 mg/dL), smoking and lack of daily exercise (exercises < 30 min/day). 

In the South-East region, which had the largest population at high risk of CKD progression (8.2%), the proportion of diabetes mellitus and hypertension are rather low. Hyper-LDL cholesterolemia and smoking are risk factors discussed in this study, but both are unlikely to cause renal injury alone. 

The Takahashi-Niimi region had a lower rate of people at high risk of CKD progression (Table 1), but the mean eGFR was the lowest among all of the regions (significantly lower than the values in the Maniwa, Tsuyama-Aida and South-West regions; Appendix A), whereas the rate of people with proteinuria was the lowest. In the Takahashi-Niimi region, the rates of people with diabetes mellitus (30.1%), diabetes mellitus at high risk of developing CKD (4.4%), hypertension (29.1%) and hyper LDL cholesterolemia (30.3%) were high (Table 2, Appendix A). In particular, the HbA1c value was significantly higher than in all other regions (Appendix A). Furthermore, the rate of people with daily exercise habit was lower, while the rate of people with poor glycemic control was higher in the Takahashi-Niimi region than in other regions. 

In contrast to the above, the Maniwa region had a significantly higher eGFR value than other regions, except for the Tsuyama-Aida region (Appendix A), and consequentially, the rate of people with an eGFR <60 mL/min was relatively low (16.9%), although the rate of people with proteinuria was high (4.3%). In the Maniwa region, the rate of people with diabetes mellitus was high (26.9%), but the rate of people with diabetes mellitus at high risk of developing CKD was low (3.2%). Furthermore, the systolic blood pressure value was significantly lower than in other regions (Appendix A), and the proportion of hypertensive people was also relatively low (21.2%, Table 2). In terms of lifestyle, the proportion of people with exercise habits was relatively low, and the average BMI was slightly high (not statistically significant), with the ratio of BMI ≥25 kg/m^2^ the highest among all regions.

## 4. Discussion

In this study, about 60–75% people were at a low risk of developing ESRD (green on CKD heatmap), while 6–9% were at a high risk (orange or red on CKD heatmap) of developing ESRD among the NHI population who underwent annual health checkups in Okayama Prefecture. However, even for regions with the same degree of severity, the causes of the CKD severity differed based on differences in the characteristics, such as the frequency of renal dysfunction or proteinuria, among the regions. This suggests that there may be differences in the underlying risk factors for CKD progression among the regions. For example, South-East region had highest proportion of severe CKD but the proportion of risk factors, such as diabetes mellites and hypertension, were rather low. Since the South-East region had the highest proportion of people with proteinuria, diseases such as glomerulonephritis were assumed as causes of CKD, in addition to diabetes mellitus and obesity; thus, increasing the health checkup rate may be effective for early detection of CKD in this region [5]. In Takahashi-Niimi region, the proportion of severe CKD was low because the proportion of people with proteinuria was low, despite low eGFR. The analysis of risk factors indicated that this region has a high proportion of individuals with diabetes mellitus, hypertension, hypercholesteremia and also lack of daily exercise. It can be thought that there are many individuals with atherosclerotic factors which cause CKD, such as nephrosclerosis [6]. The health guidance on exercise and diet, such as salt and cholesterol reduction, in the Takahashi-Niimi region. On the other hand, Maniwa region had second highest proportion of severe CKD, because of the high proportion of people with proteinuria, despite sustained eGFR. Analysis of risk factors showed that this area has a predominant proportion of diabetic mellitus and had a low ratio of hypertension. The rate of lack of exercise was high and the rate of obesity was also high. Therefore, in addition to measures for prevention of diabetes aggravation, which already have been undertaken, intervention methods such as exercise promotion, weight loss and glycemic control may be effective for prevention of CKD progression [7]. Based on these findings, it was clarified that interventions to prevent CKD development and progressions should be tailored to regional characteristics.

CKD patients were previously reported to account for 8–13% of the global population [8], and the proportions in the yellow to red categories on a CKD heatmap were reported to be 73% for yellow, 18% for orange and 9% for red [4]. In the NHI population in Okayama Prefecture, the distribution of patients categorized from yellow to red was 73–77% for yellow, 17–20% for orange and 5–8% for red, these findings were similar to previously reported data in yellow and orange [4], while the red proportion was lower than previously reported. This discrepancy may be because patients with ESRD, including those receiving chronic dialysis, tend to visit the hospital regularly and do not participate in such medical health checkups.

In the NHI health checkup system, individuals for whom a health problem has been identified receive health guidance or are recommended to visit a hospital, but the criteria vary among regions. The ratio of CKD patients at high risk of CKD progression ranged from 6–9% in each group, totaling 6498 individuals being revealed in this study. Despite differences in the number of health guidance staff and health guidance methods among municipalities, all these individuals are strongly recommended to be informed about their CKD status through the health guidance or hospital visits. However, even in cases where CKD was suspected at a medical health checkup, the number of patients who visited the hospital was reported to be approximately 2% in Japan [9] and 10% in Sweden [10]. Establishing a strategy for receiving health guidance and consultations at hospitals thus remains an unsolved issue. The lack of knowledge about CKD is also considered to be a cause of the low hospital consultation rate. A brief explanation concerning CKD that is easy for the general public to understand is thus needed. 

Diabetic nephropathy and nephrosclerosis are the highest and the second highest reasons, respectively, for starting renal replacement therapy, most patients with ESRD receive hemodialysis therapy in Japan [3]. Diabetic nephropathy is caused by diabetes mellitus, while causes of nephrosclerosis include aging, hypertension and other arteriosclerotic factors, such as dyslipidemia and smoking. These arteriosclerotic factors are also exacerbating factors for renal diseases, including diabetic nephropathy. In the present study, the prevalence of diseases that cause or exacerbate CKD was up to 20–30%, covering quite a large number of patients. We must therefore endeavor to provide health guidance to improve the health of the local population.

Only 20–30% of NHI enrollees undergo health checkups in Okayama Prefecture, which is quite low compared to the other regions in Japan. This ratio needs to be improved in the future.

This study covers the people belonging to only one of five insurances in Japan, and results are limited to one prefecture. In addition, the age of people in this study ranged 40–74 years old and their occupations were self-employed individuals, agricultural workers, fishermen, forestry workers, those who have retired from the company, those who are unemployed and those who are part-time workers who cannot take out the health insurance provided by each company; these factors may also influence their economic background. It is necessary to be careful about diversion of the result of this study to other regions, but in Japan, each insurance provides the health guidance to their users, it is rational to create the measures for prevention of CKD progression for each region based on their health checkup results.

This was a single-year study, and whether or not these results properly represent the health status of each region is unclear. It is thus necessary to accumulate survey results from several years and consider effective interventions to prevent CKD progression. 

## 5. Conclusions

Even within a single prefecture, risk factors that may exacerbate CKD progression differ, this study can provide a basis to establish an “individual” strategy for reducing CKD patient in each region.

## Figures and Tables

**Figure 1 jpm-12-00097-f001:**
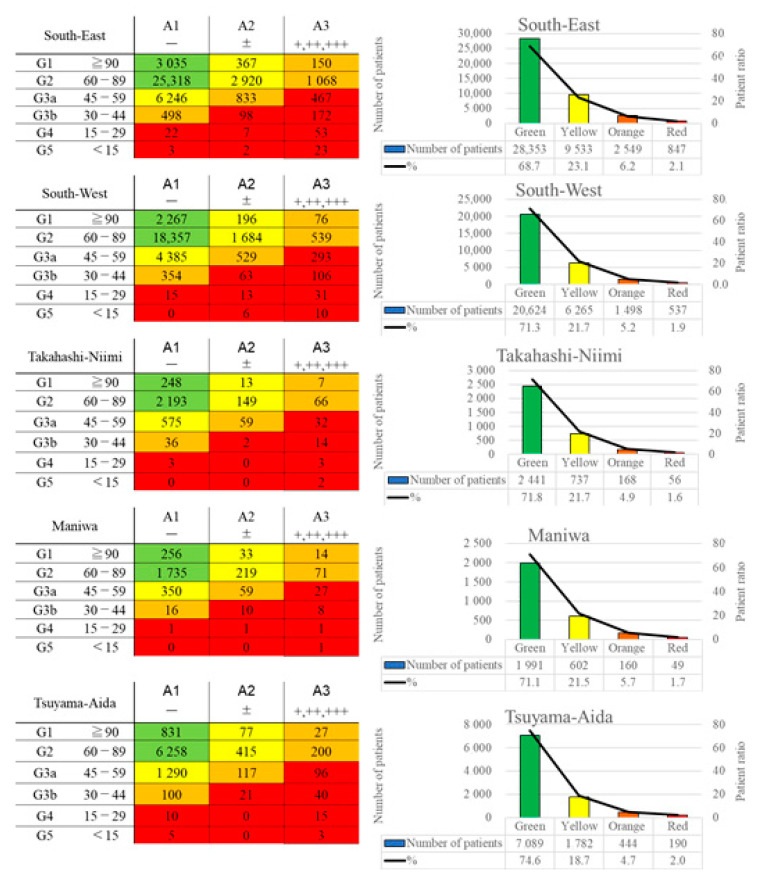
CKD heatmaps of each region. The left column shows the actual number of people in each category by the renal function and proteinuria, and the right column shows the number and proportion of people classified by risk of ESRD. ESRD: end-stage renal disease.

**Table 1 jpm-12-00097-t001:** Health checkup ratio and severity of CKD.

	South-East	South-West	Takahashi-Niimi	Maniwa	Tsuyama-Aida	*p* Value
Number of National Health Insurance users (*n*)	180,276	141,657	12,242	9850	35,935	
Number of health checkups (*n*)	42,824	29,296	3433	2914	9953	
Male/Female ratio	17,534/25,290	12,310/16,986	1532/1901	1339/1575	4518/5435	<0.001
Health checkup rate (%)	23.8	20.8	28.0	29.6	27.8	n.s.
Age (years)	66 ± 8 *	67 ± 7	67 ± 7	67 ± 7	67 ± 8	
eGFR (mL/min/1.73 m^2^)	70.8 ± 14.0	71.0 ± 14.0	70.4 ± 13.9	72.9 ± 14.0	72.2 ± 14.4	
<60 mL/min/1.73 m^2^ (*n*, %)	8430 (20.4%)	5816 (20.1%)	727 (21.4%)	474 (16.9%)	1702 (17.9%)	<0.001
Urinary Protein (+ and over) (*n*, %)	1988 (4.6%)	1069 (3.6%)	125 (3.6%)	125 (4.3%)	402 (4.0%)	n.s.
Green or Yellow on a CKD heatmap (*n*, %)	37,886 (91.8%)	26,889 (93.0%)	3178 (93.4%)	2593 (92.5%)	8871 (93.3%)	n.s.
Orange or Red on a CKD heatmap (*n*, %)	3396 (8.2%)	2035 (7.0%)	224 (6.6%)	209 (7.5%)	634 (6.7%)	n.s.

* *p* < 0.001 by a one-way analysis of variance, *p* values were obtained by a chi-square test. n.s.: not significant, eGFR: estimated glomerular filtration ratio, CKD: chronic kidney disease.

**Table 2 jpm-12-00097-t002:** Risk factors for CKD.

	South-East	South-West	Takahashi-Niimi	Maniwa	Tsuyama-Aida	*p* Value
HbA1c (%)	5.7 ± 0.6	5.7 ± 0.6	5.9 ± 0.6	5.8 ± 0.6	5.7 ± 0.7	
≥6.0% (*n*, %)	8717 (21.5%)	6467 (22.6%)	951 (30.1%)	781 (26.9%)	2051 (20.8%)	<0.001
≥7.0% (*n*, %)	1530 (3.8%)	1194 (4.2%)	139 (4.4%)	92 (3.2%)	417 (4.2%)	0.003
Systolic blood pressure (mmHg)	129.4 ± 18.0	130.0 ± 17.8	131.7 ± 18.1	127.3 ± 16.8	129.0 ± 18.0	
≥140 mmHg (*n*, %)	11,312 (26.4%)	8119 (27.7%)	1000 (29.1%)	619 (21.2%)	2476 (24.9%)	<0.001
BMI (kg/m^2^)	23.0 ± 3.6	23.0 ± 3.4	23.1 ± 3.3	23.2 ± 3.5	23.0 ± 3.6	
≥25 (%)	25.7	25.4	25.5	27.1	26.0	n.s.
LDL-cholesterol (mg/dL)	125.4 ± 30.9	123.8 ± 30.4	124.7 ± 31.0	122.1 ± 30.7	122.7 ± 33.1	
≥140 mg/dL (*n*, %)	13,146 (30.7%)	8420 (28.7%)	1041 (30.3%)	759 (26.0%)	2795 (28.1%)	<0.001
Smoking (*n*, %)	4849 (11.3%)	2788 (9.5%)	3433 (10.7%)	2914 (12.6%)	9953 (13.2%)	n.s.
Lack of daily exercise (*n*, %)	25,018 (59.2%)	15,932 (56.5%)	2100 (62.3%)	1951 (67.0%)	4570 (61.9%)	n.s.

The number and ratios of risk factors for CKD, such as diabetes mellitus (HbA1c ≥ 6.0%) and diabetes mellitus at high risk of developing CKD (HbA1c ≥ 7.0%), hypertension (systolic blood pressure ≥140 mmHg), obesity (BMI ≥ 25 kg/m^2^), hyper LDL cholesterolemia (LDL-C ≥ 140 mg/dL), smoking and lack of exercise habits. *p* values were obtained by a chi-square test, n.s.: not significant, BMI: body mass index, CKD: chronic kidney disease, LDL: low-density lipoprotein °.

## Data Availability

The datasets analyzed during the current study are not publicly available, and are requires the permission of Okayama Prefecture, because ethics committee approved this protocol only under limited condition of data availablity.

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
