# Peer review of "Prevalence of Chronic Kidney Disease and Variation of Its Risk Factors by the Regions in Okayama Prefecture"

_jpm, 2022, doi:10.3390/jpm12010097_

Round 1
Reviewer 1 Report
Introduction
-lines 38-40, suggest rephrasing this sentence to, “The medical cost associated with managing dialysis patients is about 4 million yen per year, and the increase in patients with end-stage renal disease (ESRD) is also an important issue in Japan”.
Methods
-how representative is the NHI registered cohort in comparison to the general population in Okayama Prefecture?
-in line 60, spell out what KDB stands for.
- Lines 63-64, reference has been made to the five medical regions already (in the introduction) so there is repetition.
Results
-I suggest removing this statement, ‘This section may be divided by subheadings. It should provide a concise and precise 85 description of the experimental results, their interpretation, as well as the experimental 86 conclusions that can be drawn’.
-Lines 92 to 93, suggest changing this sentence to read, “Individuals from the South-East region were significantly younger compared to other regions, but the mean difference was only one year’.
-lines 107 to 108, this sentence is discussing the results therefore it belongs to the discussion section.
-Lines 113 to 117 are similarly discussing findings and this should be done in the discussion section.
-Note as well the discussion of findings in lines 136 to 140.
-See also lines 167 to 170
Discussion
-What is the explanation for a lower eGFR in the Takahashi-Niimi Region when the region had a lower rate of people at high risk of CKD progression?
-Lines 228 to 231, these are general statements or guidance regarding this section. The authors need to discuss this in the context of their study for example, ‘Future studies should collect data over a longer duration’.
-In this cross sectional study, there were more female participants. What could be the reason for this?
-Is the socioeconomic status data for the 5 medical regions?
-The implication of the findings of this paper hinge on the development of strategies to reduce CKD progression in respective regions. To do this, the authors need to identify and explain reasons behind variations in risk factors among these regions.
Author Response
Response to Reviewer‘s Comments: JPM 1514415
Title: Prevalence of chronic kidney disease and variation of its risk factors by the regions in Okayama Prefecture
We would like to express our deep appreciation to the reviewers for their constructive comment that have assisted us in improving this manuscript. We have responded to the Reviewer’s questions and comments (italicized and bold in the Response) point-by-point below, and edited the manuscript accordingly. Changes in the revised manuscript are distinguished by track changes system in Microsoft Word. We hope this revision has improved sufficiently and is now suitable for publication in JPM.
Introduction
-lines 38-40, suggest rephrasing this sentence to, “The medical cost associated with managing dialysis patients is about 4 million yen per year, and the increase in patients with end-stage renal disease (ESRD) is also an important issue in Japan”.
In accordance with the Reviewer’s comment, we have rephased the following text in introduction section as follows;
“The medical cost associated with dialysis patients is about 4 million yen per year, and the increase in patients with end-stage renal disease (ESRD) is also an important issue in Japan.”
“The medical cost associated with managing dialysis patients is about 4 million yen per year, and the increase in patients with end-stage renal disease (ESRD) is also an important issue in Japan.”
Methods
-how representative is the NHI registered cohort in comparison to the general population in Okayama Prefecture?
National Health Insurance in Okayama Prefecture covers 20.1% of population, who are under 75 years old and are self-employed individuals, agricultural workers, fishermen, forestry workers, those who have retired from the company, those who are unemployed, and those who are part-time workers who cannot take out the health insurance provided by each company. The age of people who underwent the NHI’s health checkup ranges 40-74 years old. Therefore, the result of this study cannot be regarded as a precise, representative value of Okayama Prefecture, but it is enough to consider interventions for prevention of CKD progression for subjects who are covered by NHI.
-in line 60, spell out what KDB stands for.
In accordance with the reviewer's comment, we spelled it out as follows:
the NIH database (Kokuho Data Base, KDB) system developed by the All-Japan Federation of NHI Organizations.
- Lines 63-64, reference has been made to the five medical regions already (in the introduction) so there is repetition.
Thank you so much. We corrected this error in accordance with the reviewer's comment.
Results
-I suggest removing this statement, ‘This section may be divided by subheadings. It should provide a concise and precise 85 description of the experimental results, their interpretation, as well as the experimental 86 conclusions that can be drawn’.
I apologize for making such a mistake. We deleted the text in accordance with the reviewer’s comment.
-Lines 92 to 93, suggest changing this sentence to read, “Individuals from the South-East region were significantly younger compared to other regions, but the mean difference was only one year’.
We would like to thank to the reviewer’s comment, we have changed this text as suggested by the reviewer. It became a clearer sentence.
-lines 107 to 108, this sentence is discussing the results therefore it belongs to the discussion section.
-Lines 113 to 117 are similarly discussing findings and this should be done in the discussion section.
-Note as well the discussion of findings in lines 136 to 140.
-See also lines 167 to 170
In accordance with the reviewer’s comment on these problems, we deleted these text and also lines 154-155 in Result and added new paragraph in Discussion as following:
In this study, about 60%-75% people were at a low risk of developing ESRD (green on CKD heatmap), while 6%-9% were at a high risk (orange or red on CKD heatmap) of developing ESRD among the NHI population who underwent annual health checkups in Okayama Prefecture. However even for regions with the same degree of severity, the causes of the CKD severity differed based on differences in the characteristics, such as the frequency of renal dysfunction or proteinuria, among the regions. This suggests that there may be differences in the underlying risk factors for CKD progression among the regions. For example, South-East region had highest proportion of severe CKD but the propor-tion of risk factors, such as diabetes mellites and hypertension, were rather low. Since the South-East region had the highest proportion of people with proteinuria, diseases such as glomerulonephritis were assumed as causes of CKD, in addition to diabetes mellitus and obesity, thus, increasing the health checkup rate may be effective for early detection of CKD in this region[5]. In Takahashi-Niimi region, the proportion of se-vere CKD was low because the proportion of people with proteinuria was low, despite of low eGFR. The analysis of risk factors indicating that this region has high propor-tion of individuals with diabetes mellitus, hypertension, hypercholesteremia, and also lack of daily exercise. It can be thought that there are many individuals with athero-sclerotic factors, which causes CKD such as nephrosclerosis [6]. The health guidance on exercise and diet, such as salt and cholesterol reduction, in the Takahashi-Niimi region. On the other hand, Maniwa region had second highest proportion of severe CKD, be-cause of the high proportion of people with proteinuria, despite of sustained eGFR. Analysis of risk factors showed that this area has predominant proportion of diabetic mellitus and had low ratio of hypertension. The rate of lack of exercise was high and the rate of obesity was also high. Therefore, in addition to measures for prevention of diabetes aggravation, which already have been undertaken, intervention methods such as exercise promotion, weight loss and glycemic control may be effective for preven-tion of CKD progression. This cross-sectional study on risk factors suggested that different methods for preventing CKD development and progression might be required for different regions; for example, an increase in the rate of annual health checkups might be more appropriate in the South-East area, while the promotion of community-based exercise and dietary guid-ance targeting diabetes, hypertension, dyslipidemia or obesity might be more appropriate in other regions. Based on these findings, it was clarified that interventions to prevent CKD development and progressions should be tailored to regional characteristics.
Discussion
-What is the explanation for a lower eGFR in the Takahashi-Niimi Region when the region had a lower rate of people at high risk of CKD progression?
We would like to appreciate for the reviewer's comment on this point. The reason why the rate of people at high risk of CKD progression is low in Takahashi-Niimi region is few people had proteinuria. On the other hand, the reason why eGFR is low in this region cannot precisely confirm, but we think it should be considered from the risk factors of CKD. In Takahashi-Niimi region, the rate of people with diabetes mellites and hypertension is highest among 5 regions. In addition, the ratio of people with hypercholesteremia is second highest, and it is thought that there are many individuals with atherosclerotic factors, which causes nephrosclerosis.
-Lines 228 to 231, these are general statements or guidance regarding this section. The authors need to discuss this in the context of their study for example, ‘Future studies should collect data over a longer duration’.
I apologize for making such a mistake. The text of the template did not have been deleted. We deleted the text in accordance with the reviewer’s comment.
-In this cross sectional study, there were more female participants. What could be the reason for this?
We thank the reviewer for this comment. Considering that NHI covers individuals who have retired from the company, those who are unemployed, and those who are part-time workers who cannot take out the health insurance provided by each company, the low employment rate of women (male 84% vs female 70% , data from (The White Paper on Gender Equality 2020, Gender Equality Bureau Cabiner Office in Japan) in Japan could have led to an increased rate of female NHI users. Moreover, female population is slightly larger in Japan.
-Is the socioeconomic status data for the 5 medical regions?
We apologize for not having actual data on socio-economic status in this Prefecture, but the three northern regions of Okayama Prefecture (Takahashi-Niimi, Maniwa, Tsuyama-Aida) are highlands and thriving in agriculture and dairy. On the other hand, the southern part is an urban area where the population is concentrated, and it also has an industrial area.
-The implication of the findings of this paper hinge on the development of strategies to reduce CKD progression in respective regions. To do this, the authors need to identify and explain reasons behind variations in risk factors among these regions.
We would greatly appreciate for the reviewer’s comment. Since this study deals only the results of health checkup, we can only mention that the components of CKD differ from region to region, therefore the prevention measures for CKD progression must to be considered tailored to each region. In addition, although it is not yet in preliminary research, we are collecting data on the annual change in number and reason of dialysis patients and patients who introduced dialysis, and healthcare cost of dialysis patients and CKD patients, and reports the data to each region for their health promotion.
Reviewer 2 Report
I would like to think the authors for their work on this important topic and for the opportunity to review. I believe the premise of the manuscript that regionally different co-morbid conditions are driving the potential for ESRD. While I feel like the paper is interesting, I feel that there needs to be major modifications to the analytical methods prior to drawing the conclusions that the authors did.
Major
- More discussion is need regarding the potential impact selection bias may have on the results.
- The average eGFR values seem extremely low compared to what we would expect in a healthy population. More discussion about this is warranted.
- It appears that the authors are just basing their conclusions off the comparison of percentages by region. An individual logistic regression examining each of these risk factors with stratification by region to determine differences would be a more appropriate approach to draw conclusions on.
- While figure 2 is pretty, it does not really support evidence that there are differences between regions. Just looking at the graph my thought would be there is no difference. Running a logistic regression would allow you to plot the OR and confidence intervals for each by region which may provide a better graphical representation of the differences.
- There are only 7 previous manuscripts cited. There has been loads of work done examining different co-morbid risk factors for CKD and ESRD. I would advise that the author’s review the literature and incorporate more of these studies into their introduction and discussion.
Minor
- For readers unfamiliar with the geography of Japan a visual representation of what is described at line 46 would be helpful.
- Lines 85-87 can be removed
- Line 113 is better suited for the discussion section
- Figure 1, right panel would be better displayed as a bar chart.
- Lines 167-170 are better suited for the discussion section
Author Response
Response to Reviewer‘s Comments: JPM 1514415
Title: Prevalence of chronic kidney disease and variation of its risk factors by the regions in Okayama Prefecture
We thank the Reviewers for their constructive comments that have assisted us in improving this manuscript. We have responded to the Reviewers’ questions and comments (italicized in the Response) point-by-point below, and edited the manuscript accordingly. Changes in the revised manuscript are distinguished in track changes system in Microsoft Word. We hope this revision has improved sufficiently and is now suitable for publication in JPM.
Major
- More discussion is need regarding the potential impact selection bias may have on the results.
We agree that this point requires clarification, and have added the following text in Discussion;
This study covers the people belonging to only one of five insurances in Japan, and result of limiting it to one prefecture. In addition, the age of people in this study ranged 40-74 years old and their occupations are self-employed individuals, agricultural workers, fishermen, forestry workers, those who have retired from the company, those who are unemployed, and those who are part-time workers who cannot take out the health insurance provided by each company, these factors may also influence their economic background. It is necessary to be careful about diversion of the result of this study to other region, but in Japan, each insurance provides the health guidance to their users, it is rational to create the measures for prevention of CKD progression for each region based on their health checkup results.
- The average eGFR values seem extremely low compared to what we would expect in a healthy population. More discussion about this is warranted.
We agree that the average eGFR values low compared to that of health population, this point requires clarification.
The mean eGFR of Japanese people is reported 72.7±17.0 ml/min/1.73m2 in male and 75.0±18.0 ml/min/1.73m2 in female from Japanese Government Statistics in 2018, and their average age is 44 years old. Estimated GFR slowly declines 0.36ml/min/1.73m2/year in Japanese people (Imai Hypertension Res 2008), thus even if the value of creatinine is normal range, eGFR is distributed 58.5-102.4ml/min/1.73m2 in male and 43.3-75.7ml/min/1.73m2 in female at age 65 (CKD guide 2012). The distribution of age of all subject is shown in supplemental figure1, and the mean age in this study is 67.8±7 (table1). According to Japanese Government Statistics in 2018, the mean eGFR at 60-69 years old are 71±14.2 ml/min/1.73m2 in male and 71.4±13.7 ml/min/1.73m2 in female. The reason why the lower eGFR in this study is that the subjects of this study are limited to 40-74 years old, and contains more elder people.
- It appears that the authors are just basing their conclusions off the comparison of percentages by region. An individual logistic regression examining each of these risk factors with stratification by region to determine differences would be a more appropriate approach to draw conclusions on.
We appreciate the reviewer’s instruction for performing logistic regression analysis on risk factors. We examined the association of each risk factors and region. We used multivariate logistic regression to control for the potentially confounding roles of age, sex, eGFR. The results are listed below;
|
South-East |
South-West |
Takahashi-Niimi |
Maniwa |
Tsuyama-Aida |
HbA1c ≥ 6.0% |
0.918 [0.888-0.949] * |
1.028 [0.993-1.064] |
1.404 [1.299-1.517] * |
1.333 [1.223-1.454] * |
0.913 [0.865-0.963] * |
HbA1c ≥7.0% |
0.933 [0.869-1.001] |
1.086 [1.009-1.168] * |
1.018 [0.853-1.215] |
0.785 [0.634-0.972] * |
1.051 [0.943-1.171] |
Systolic blood pressure ≥140 mmHg |
1.018 [0.988-1.050] |
1.047 [1.014-1.081] * |
1.112 [1.030-1.200] * |
0.729 [0.665-0.799] * |
0.905 [0.861-0.951] * |
BMI ≥25 kg/m2 |
1.001 [0.970-1.032] |
0.984 [0.953-1.017] |
0.974 [0.900-1.054] |
1.105 [1.015-1.204] * |
1.011 [0.962-1.061] |
LDL-cholesterol ≥140 mg/dl |
1.096 [1.064-1.129] * |
0.940 [0.912-0.970] * |
1.045 [0.970-1.127] |
0.841 [0.772-0.917] * |
0.943 [0.899-0.988] * |
Smoking |
1.092 [1.043-1.142] * |
0.824 [0.784-0.865] * |
0.960 [0.854-1.078] |
1.071 [0.949-1.209] |
1.196 [1.118-1.280]* |
Lack of daily exercise |
1.726 [1.646-1.810] * |
0.857 [0.832-0.882] * |
1.146 [1.067-1.231] * |
1.341 [1.237-1.454] * |
1.726 [1.646-1.810] * |
*P<0.05
This result clarifies the contribution of each risk factor by region, but this method does not reveal the number of people to be intervened in each region. Thus, we decided not to change the text because there is no contradiction between the risk factors obtained from the logistic analysis and the results of the examination based on the ratios we conducted in the original text.
While figure 2 is pretty, it does not really support evidence that there are differences between regions. Just looking at the graph my thought would be there is no difference. Running a logistic regression would allow you to plot the OR and confidence intervals for each by region which may provide a better graphical representation of the differences.
We agree with that figure2 gives confusion. Because figure2 shows the same data (rate of each risk factors) in table2. So, we deleted figure2, along with this, we also deleted the following text in result;
Interestingly, in Figure 2, the regions with a high proportion of people with daily ex-exercise habits tended to have lower rates of diabetes mellitus, whereas the regions with a high proportion of people without daily exercise habits tended to have higher rates of diabetes mellitus, although the relationship between exercise habit and BMI was un-clear.
- There are only 7 previous manuscripts cited. There has been loads of work done examining different co-morbid risk factors for CKD and ESRD. I would advise that the author’s review the literature and incorporate more of these studies into their introduction and discussion.
In accordance with the reviewer’s comment, we added some previous manuscript in introduction and discussion.
Minor
- For readers unfamiliar with the geography of Japan a visual representation of what is described at line 46 would be helpful.
We agree that additional information on regional characteristics of Okayama Prefecture is needed for people who are not familiar with Japan. Okayama is located at the western part of Japan, next to Hiroshima prefecture.
- Lines 85-87 can be removed
I apologize for making such a mistake. We deleted the text in accordance with the reviewer’s comment.
- Line 113 is better suited for the discussion section
Lines 167-170 are better suited for the discussion section
In accordance with the reviewer’s comment on these problems, we deleted these text in Result and added new paragraph in Discussion as following:
In this study, about 60%-75% people were at a low risk of developing ESRD (green on CKD heatmap), while 6%-9% were at a high risk (orange or red on CKD heatmap) of developing ESRD among the NHI population who underwent annual health checkups in Okayama Prefecture. However even for regions with the same degree of severity, the causes of the CKD severity differed based on differences in the characteristics, such as the frequency of renal dysfunction or proteinuria, among the regions. This suggests that there may be differences in the underlying risk factors for CKD progression among the regions. For example, South-East region had highest proportion of severe CKD but the propor-tion of risk factors, such as diabetes mellites and hypertension, were rather low. Since the South-East region had the highest proportion of people with proteinuria, diseases such as glomerulonephritis were assumed as causes of CKD, in addition to diabetes mellitus and obesity, thus, increasing the health checkup rate may be effective for early detection of CKD in this region[5]. In Takahashi-Niimi region, the proportion of se-vere CKD was low because the proportion of people with proteinuria was low, despite of low eGFR. The analysis of risk factors indicating that this region has high propor-tion of individuals with diabetes mellitus, hypertension, hypercholesteremia, and also lack of daily exercise. It can be thought that there are many individuals with athero-sclerotic factors, which causes CKD such as nephrosclerosis[6]. The health guidance on exercise and diet, such as salt and cholesterol reduction, in the Takahashi-Niimi region. On the other hand, Maniwa region had second highest proportion of severe CKD, be-cause of the high proportion of people with proteinuria, despite of sustained eGFR. Analysis of risk factors showed that this area has predominant proportion of diabetic mellitus and had low ratio of hypertension. The rate of lack of exercise was high and the rate of obesity was also high. Therefore, in addition to measures for prevention of diabetes aggravation, which already have been undertaken, intervention methods such as exercise promotion, weight loss and glycemic control may be effective for preven-tion of CKD progression[7]. This cross-sectional study on risk factors suggested that different methods for preventing CKD development and progression might be required for different regions; for example, an increase in the rate of annual health checkups might be more appropriate in the South-East area, while the promotion of community-based exercise and dietary guidance targeting diabetes, hypertension, dyslipidemia or obesity might be more appropriate in other regions. Based on these findings, it was clarified that interventions to prevent CKD development and progressions should be tailored to regional characteristics.
- Figure 1, right panel would be better displayed as a bar chart.
In accordance with reviewer’s comment, we changed the right panel in figure1 to bar chart.
Reviewer 3 Report
In their manuscript Umebayashi et. al. try to correlate the prevalence of chronic kidney disease with the variation of different risk factors in different regions.
In my opinion there are different limitations in this study. CKD itself is a very multifactorial disease which can not be linked to one risk factor. Another critical factor is the short evaluation period of just one year and the low percentage of participants in the health check ups. I do not think that this is a representitive proportion to draw a conclusion of the health status in a defined region. In general differences between the different regions are very small and it is hard to say if there are really relevant. Authors also do not approach potential reasons for this differences.
I think more data should be accumulated and should be interpreted and discussed in the context of previous studies.
Author Response
Response to Reviewer‘s Comments: JPM 1514415
Title: Prevalence of chronic kidney disease and variation of its risk factors by the regions in Okayama Prefecture
We thank the Reviewers for their constructive comments that have assisted us in improving this manuscript. We have responded to the Reviewers’ questions and comments (italicized in the Response) point-by-point below, and edited the manuscript accordingly. Changes in the revised manuscript are distinguished in track changes system in Microsoft Word. We hope this revision has improved sufficiently and is now suitable for publication in JPM
In my opinion there are different limitations in this study.
- CKD itself is a very multifactorial disease which can not be linked to one risk factor.
We agree with the reviewer’s comment that CKD is a multifactorial disease, that cannot be simply correspond one-to-one with each risk factor, such as proteinuria corresponding to diabetic nephropathy. In addition to risk factors we mentioned in introduction, other risk factors such as drug usage history, surgery history and uric acid level are also important. In past decade, diabetic nephropathy was a main cause of ESRD, the measures to prevention of diabetic nephropathy were taken, but at present, end-stage renal disease caused by other disease, such as nephrosclerosis is increasing. Therefore, interventions for other factors are needed. As reviewer indicates, CKD, especially the currently increasing nephrosclerosis, is caused by many factors, but it is difficult to provide intervention actions decreasing hypertension, hypercholesteremia, smoking, and obesity at the same time. Thus, we are trying to control risk factors by determining the importance of each risk factors according to the characteristics of the region. To clarifying this point, we added the text in introduction as follows;
Diabetic nephropathy is the most common cause of ESRD, thus various preventive measures against diabetic nephropathy have been taken and the number of dialysis introduction due to diabetic nephropathy has begun to decrease in Japan[3]. On the other hand, the number of patients who started renal replacement therapy is increasing year by year, therefore, the comprehensive strategies for preventing CKD development and progression are required.
- Another critical factor is the short evaluation period of just one year and the low percentage of participants in the health check ups. I do not think that this is a representitive proportion to draw a conclusion of the health status in a defined region. In general differences between the different regions are very small and it is hard to say if there are really relevant. Authors also do not approach potential reasons for this differences.
We agree with the reviewer comment that the small number of people surveyed is a major limitation of this study. In addition, this study included data on age ranged 40-74 and only one insurance. These points are well taken in this kind of the observational study. However, each insurer provides health guidance under the Japanese insurance system, so, the data from each insurance group is sufficient to make a measure for preventing CKD development and progression. In addition, we have accumulated data on health checkup results, medical costs, and dialysis patients over the past three years, but there has been no significant change in the tendency of health checkup data. Therefore, we decided to report our findings in this single-year cross-sectional study.
I think more data should be accumulated and should be interpreted and discussed in the context of previous studies.
We would like to thank the reviewer for this thoughtful comment to improve our manuscript. In accordance with the reviewer’s comment, we added some literature in introduction and discussion.
Round 2
Reviewer 1 Report
The authors have addressed my comments comprehensively. The paper still requires moderate English editing.
Best wishes
Reviewer 3 Report
The authors responded appropriately to the concerns and improved there manuscript accordingly. There are still limitations to the study, but it is now much more suitable for publication.